# Application of GIS in Agriculture in Promoting Evidence-Informed Decision Making for Improving Agriculture Sustainability: A Systematic Review

Mwehe Mathenge [1,*], Ben G. J. S. Sonneveld [2] and Jacqueline E. W. Broerse [2]

[1] Department of Urban Management, School of Planning and Architecture, Maseno University, Maseno 40105, Kenya
[2] Athena Institute, Faculty of Science, Vrije Universiteit Amsterdam, De Boelelaan 1085, 1081 HV Amsterdam, The Netherlands
[*] Correspondence: mmwehe@maseno.ac.ke; Tel.: +254-7243-02883

**Abstract:** The objective of this review was to synthesize existing evidence on GIS and RS application in agriculture in enhancing evidence-informed policy and practice for improving agriculture sustainability and identifying obstacles to their application, particularly in low- and middle-income countries. Systematic searches were conducted in the databases SCOPUS, Web of Science, Bielefeld Academic Search Engine, COnnecting REpositories (CORE), and Google Scholar. We identified 2113 articles published between 2010–2021, out of which 40 articles met the inclusion criteria. The results show that GIS technology application in agriculture has gained prominence in the last decade, with 66% of selected papers being published in the last six years. The main GIS application areas identified included: crop yield estimation, soil fertility assessment, cropping patterns monitoring, drought assessment, pest and crop disease detection and management, precision agriculture, and fertilizer and weed management. GIS technology has the potential to enhance agriculture sustainability through integrating the spatial dimension of agriculture into agriculture policies. In addition, GIS potential in promoting evidenced informed decision making is growing. There is, however, a big gap in GIS application in sub-Saharan Africa, with only one paper originating from this region. With the growing threat of climate change to agriculture and food security, there is an increased need for the integration of GIS in policy and decision making in improving agriculture sustainability.

**Keywords:** GIS; RS; spatial autocorrelation; policy integration; agri-spatial policy integration; spatial; sustainable agri-food systems

## 1. Introduction

The demand for food globally has risen tremendously and is expected to increase to 59–98% by the year 2050 [1]. However, the growing concerns are that the agricultural food production systems are unable to match the high demand, especially in poor nations, causing an intensifying level of food insecurities [2]. The inefficiency of the food production systems is also one of the reasons for food insecurity [3]. How best to facilitate increased food production without jeopardizing land and water resources, energy, and the environment is a momentous task that government and policymakers have to address [4].

In many low- and middle-income countries (LMICs), most of the food production is rural-based, dominated by smallholder and subsistence farmers. Enhancing smallholders' sustainability requires that farmers are empowered with practicable information that enables them not only to make evidence-informed decisions but also to implement them in activities that could increase their farm productivity and sustainability. In efforts toward transforming the weak and often inefficient traditional subsistence production practices, sustainable production approaches [5,6] that support production-efficiency-enhancement and better agronomic practices are needed. These include planting climate-resilient crops,

high-yielding crop varieties, crop yield forecasting, integrated pest management, as well as integrating biodiversity solutions in sustainable food production systems [7,8] Ultimately, these novel interventions would require comprehensive, up-to-date datasets (spatial and non-spatial) and the adoption of advanced GIS technologies that can synthesize and integrate social, spatial, economic, demographic, and environmental data in agriculture. The output of this synthesis would be evidence-based spatial knowledge that improves our understanding of agriculture sustainability and in supporting better policies and decision-making processes.

Contemporary advances in Geographic Information Systems (GIS), Remote Sensing (RS), and Geographic Positioning Systems (GPS) technologies present an opportunity to acquire and operationalize high-resolution satellite imagery and digital spatial data [9]. In the agriculture sector, these spatial data have aided in the investigation of the spatial linkages of social, physical, agroecological, and environmental complexities and how they affect agriculture sustainability. GIS technology provides users with a mixture of geo-spatial information management tools and methods that allow users to collect, store, integrate, query, display, and analyze geospatial data at various scales [10]. Remote sensing technology acquire images and other information about crops and soil from sensors mounted on different platforms including satellites, airborne remote sensing (manned drones and unmanned aerial vehicles), and ground-based equipment that is then processed by computers to aid agricultural decision-making systems [11,12].

The spatial context of agriculture can be viewed from the perspective of farmers' differentiated access to livelihood capitals, local resources, and access to essential infrastructure and services existing in a locality. In a GIS system, the data containing each of these aspects can be deconstructed as nested spatial layers, each rooted in local geography by geographic coordinates captured using GPS [13]. These spatial layers can then be processed and analyzed in a GIS system in multiple ways to reveal crop and soil conditions and spatial interactions, predict crop trends, monitor land-use change, monitor pests, and in biodiversity conservation [14–17]. They can also be used to map and reveal spatial impediments to agricultural production, or even new information for improving agricultural sustainability.

In recent times, the increasing complexity associated with agriculture production systems has aroused policymakers' interest in investigating how the spatial aspect "dimension" of agriculture can be exploited using advanced GIS, RS, and GPS technologies to improve agricultural productivity and better production practices [18,19]. The integration of GIS technologies in agriculture has increased the opportunities for the development of even better spatial explicit frameworks that support the creation of dynamic agriculture databases and interactive systems [20]. Such database systems allow users to interact with spatially referenced agriculture data in real-time, while accurately providing precise positional data, thus providing enhanced frameworks for decision making. New fields that apply GIS in agriculture have emerged as a result. These include precision agriculture, automated farm systems, crop yield forecasting, climate change detections, and the real-time monitoring of crop production [11,12,21]. These have the capability of improving agricultural production and food security.

In this regard, several recent systematic literature reviews have been conducted to illuminate and consolidate various ways GIS, RS, and GPS technologies have been applied in the agriculture sector. García-Berná et al. [11] used a systematic mapping study to focus on the current trend and what new opportunities in remote sensing techniques offer in agriculture. Their study found increased uptake of RS technologies in the acquisition and extracting of georeferenced data from satellite imagery and unmanned aerial vehicles. Spatial data from these technologies have been applied in several areas including crop growth and yield estimation, cropland parameter extraction, weed and disease detection, and the monitoring of water and nutrients in plants. How this application could be integrated to improve spatial-based agriculture policymaking was not elaborated by the authors. The Al-Ismaili [22] review highlighted the integrated application of RS and GIS

techniques in precision agriculture and in the mapping, detection, and classification of the greenhouse through aerial images and satellites. How such a technique could be assimilated into enhancing policymaking was not mentioned. In yet another meta-review, Weiss et al. [12] research highlighted the emerging development in RS that strengthens the specific application of RS in crop breeding, agricultural land use monitoring, crop yield forecasting, and biodiversity loss. Sharma, Kamble, and Gunasekaran [23] focused on how GIS data applications have assisted in the development of precision agriculture. The authors proposed a framework, "Big GIS Analytic", to guide how big GIS data should be applied in the agriculture supply chain. Their framework also lays a foundation for a theoretical structure for improving the quality of GIS data application in agriculture to elevate productivity. These studies help us to understand how GIS and RS applications in agricultural production systems have advanced. However, the available systematic reviews seem not to explicitly provide how GIS and RS technologies could enhance the integration of the spatial dimension of agriculture into policy frameworks and interventions.

There is an increasing demand for evidence-supported decision making to assist policymakers in assessing the local needs of farmers, improving production and supply value chains, and developing spatial based-interventions. In this regard, this review aimed at synthesizing existing evidence on GIS and RS application in agriculture in enhancing evidence-informed policies for improving agriculture sustainability and identifying obstacles for their application, particularly in LMICs. The review draws on a decade of literature, from 2011 to 2021, to examine the current and future perspectives on integrating GIS in policies that support agriculture sustainability. The main contributions of the study are to provide readers and policymakers with evidence on how GIS technology has been used in the agriculture sector to improve agricultural production practices and inform how the technology can be adopted to improve evidence-based decision making and policies. This paper is structured as follows: after the introduction, we describe the methodology applied to select and review the articles; then, we detail the findings in Section 3. In Section 4, we highlight obstacles to applying GIS in agriculture policy and practice. Lastly, Sections 5 and 6 give the conclusion and the limitations of the study.

## 2. Review Methodology

### 2.1. Process of Screening

The search used the bibliographic databases SCOPUS, Web of Science/Clarivate, Bielefeld Academic Search Engine (BASE), and COnnecting REpositories (CORE), as well as Google Scholar. The following inclusion criteria were employed to screen for titles and abstracts: (1) full articles in peer-reviewed journals; (2) articles published between January 2010 and October 2021; (3) written in the English language; and (4) those associated with the application of GIS or RS in agriculture. The following search string was applied as index terms to search: *"TITLE, ABSTRACT (Agriculture\* OR Plant OR Crop\*) AND (GIS OR Geographic OR Information OR Systems) AND (Remote OR Sensing OR RS)"*. The full search syntax is found in Table A1 in the Appendix A. Following the eligibility criteria, a total of 2113 articles were found; 701 articles were identified from SCOPUS, 104 from Web of Science, 468 from Bielefeld Academic Search Engine (BASE), 68 from CORE, and 238 records from Google Scholar. After excluding duplicates and studies for which no full text or access was available (988 articles), 1223 articles were eligible for further screening.

The flow of the screening process is shown in Figure A1 in the Appendix A. The first screening was based on the title and abstract checking for relevance to the purpose of this article, based on which a substantial number of articles ($n = 554$) were excluded. Further exclusion criteria were based on (1) articles focusing on the general application of GIS, i.e., suitability analysis and site selection analysis ($n = 81$) and (2) irrelevant topics or focus ($n = 171$). After the exclusion of these articles, 97 articles were subjected to secondary screening through full article reading, which resulted in the exclusion of 57 articles. After the final screening, 40 articles were selected for the analysis.

### 2.2. Data Extraction and Analysis

Full reference records for selected articles were exported to the Mendeley reference manager and Microsoft Excel to enable coding and analysis. We extracted data using a standardized form and included the following descriptive data: author(s); year of study; journal; location; research objectives/questions; and main methods, findings, and conclusions. The included articles were analyzed through thematic analysis, combining both deductive and inductive coding.

## 3. Results

### 3.1. Characterization of the Selected Papers

In total, 20 journals published the selected papers (Figure 1), with the top four journals being Elsevier (26% of the articles), Springer (21%), MDPI (9%), and PLOS ONE (9%). Affiliate journals of Elsevier where the papers were published included Agricultural Systems, Chemosphere, Science of the Total Environment, Agricultural Water Management, Field Crops Research, Computers and Electronics in Agriculture, Applied Geography, Computers and Electronics in Agriculture, and Catena. The Springer journal affiliates included Nature, Earth Systems and Environment, Nutrient Cycle Agroecosystem, Precision Agriculture, and Environment Monitoring Assess, while the MDPI affiliate journals included Sustainability and Agriculture.

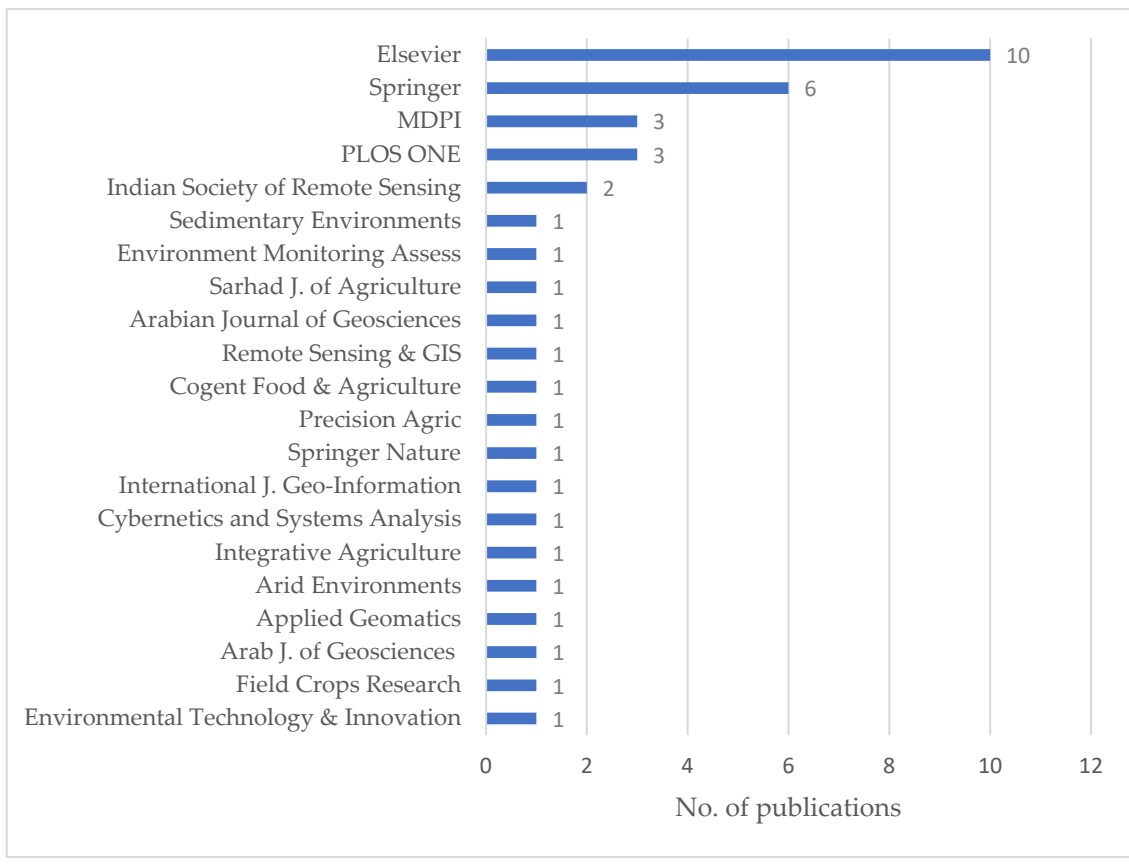

**Figure 1.** Publication sources of selected papers.

The selected articles covered diverse fields of GIS applications that were published in various years and based in diverse regions as shown in Table 1.

**Table 1.** Characteristics of the included records.

| Criteria | Category | No. of Articles | % |
|---|---|---|---|
| Field of GIS application | Crop yield estimation/forecasting | 12 | 30% |
| | Soil fertility assessment | 9 | 22.5% |
| | Cropping pattern and monitoring | 4 | 10% |
| | Drought risk assessment | 3 | 10% |
| | Pest and crop disease detection | 3 | 7.5% |
| | Precision agriculture | 3 | 7.5% |
| | Fertilizer and weed management | 2 | 5% |
| Publication year |  | | |
| Region of case study | East Asia and pacific | 14 | 35% |
| | Europe and Central Asia | 3 | 7.5% |
| | South Asia | 11 | 27.5% |
| | North America | 4 | 10% |
| | The Middle East and North Africa | 7 | 17.5% |
| | Sub-Sahara Africa | 1 | 2.5% |

The most frequent fields of application were crop yield estimation and forecasting (30%) and soil fertility assessment (22.5%). Eighteen countries were identified in the selected papers where the research was conducted. Grouped by region, we found that East Asia and Pacific countries were the most frequent, accounting for 35% of the total, including Australia ($n = 4$); Bangladesh ($n = 1$); Indonesia ($n = 1$); China ($n = 7$); and Russia ($n = 1$). South Asia accounted for 27.5% including India, ($n = 7$); Pakistan, ($n = 1$); and Iran ($n = 3$). North America accounted for 10% of the total, including the USA ($n = 3$) and Canada ($n = 1$). Middle East and North Africa accounted for 17.5% including Saudi Arabia, ($n = 2$); UAE, ($n = 1$); Morocco, ($n = 1$); and Egypt, ($n = 3$). GIS applications in Europe and Central Asia accounted for 7.5% of the total, including Ireland, Ukraine, and Turkey, each with ($n = 1$). Sub-Saharan Africa had the least articles, with only one (Ethiopia, $n = 1$) accounting for 2.5%.

The most frequent type of GIS application methodologies identified in the selected papers are presented in Figure 2. More than half (27 papers) accounting for 67.5% of the selected papers used GIS in their methodologies; 8 papers (20%) integrated both GIS and RS, while 5 papers (12.5%) used RS techniques.

*3.2. GIS Application in Agriculture and the Implication to Policy*

The main field of study for the selected papers was categorized into seven application areas (Table 2). These include crop yield estimation/forecasting (26% of the papers), soil fertility assessment (18%), cropping patterns and agricultural monitoring (13%), drought assessment (16%), pest and crop disease detection and management (11%), precision agriculture (8%), and fertilizer and weed management (8%).

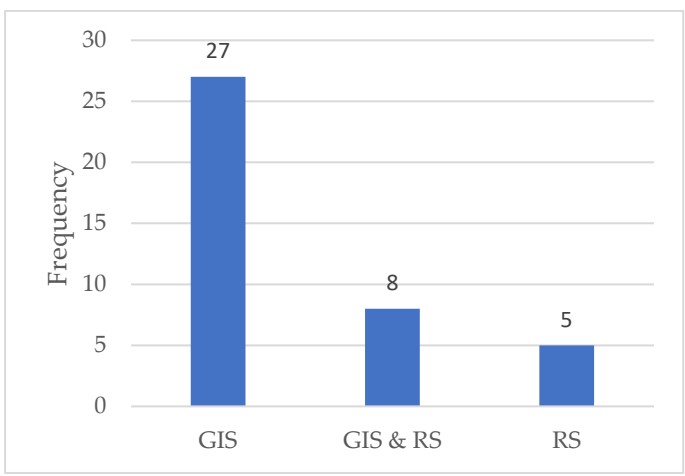

**Figure 2.** Number of papers using GIS, RS, or a combination of the two in their methodology.

**Table 2.** Classification of main types of research topics addressed in the selected papers.

| | Research Topic/GIS Application Areas | Reference | No. of Publications |
|---|---|---|---|
| 1. | Crop yield estimation/Forecasting | [24–31] | 12 |
| 2. | Soil fertility assessment | [32–38] | 7 |
| 3. | Cropping patterns and agricultural monitoring | [16,39–42] | 5 |
| 4. | Drought assessment | [43–48] | 6 |
| 5. | Pest and crop disease detection and management | [49–52] | 4 |
| 6. | Precision agriculture | [53–55] | 3 |
| 7. | Fertilizer and weed management | [21,56,57] | 3 |

We expound on how GIS was applied in the selected papers according to the research topic in the sections below.

### 3.2.1. Crop Yield Estimation/Forecasting

Monitoring crop growth and early crop yield forecasting over agricultural fields is an important procedure for food security planning and agricultural economic return prediction. The continued advancement in RS and GIS technologies has improved the process and techniques of monitoring the development of crops and estimating their yields [26,29,31]. Several studies demonstrate the application of integrated GIS and RS technologies in crop yield estimation. Memon et al.'s [24] study demonstrated how integrating multispectral Landsat satellite imagery and comparing different RS-based spectral indices were effective in measuring the percentage of wheat straw cover and successively determining its effect on yields of rice crops. The knowledge can inform long-term planning of agriculture sustainability in rice-wheat cropping systems. The result of the research by Hassan and Goheer [27] showed that the accurate early estimation of wheat crop yield before harvesting can be determined by using vegetation indices derived from moderate resolution imaging spectroradiometer satellite imagery and crop yield data and the GIS modelling approach. In yet another study, Hassan and Goheer [28] used a GIS-based environment policy integrated climate model that provided a practical tool for simulating rice crop yield. The model combined regional level crop level data, soil data, farm management data, and climatic data to spatially estimate variations in crop yield. Likewise, Al-Gaadi et al. [29] extracted the normalized difference vegetation index and soil-adjusted vegetation index from Landsat satellite images acquired during the potato growth stages to predict potato tuber crop yield. GIS- and RS-based crop yield forecasting models could have a wider application in informing spatially based agriculture policies. For example, based on the output of these models, policy intervention can be designed to manipulate the specific contributors to crop

yields (which include farm management techniques, weather conditions, water availability, altitude, terrain, plant health, and policy intervention) [25,30]. Forecasting crop yields well before harvest is crucial, especially in a region characterized by climatic uncertainties. Monitoring agricultural crop growth conditions and the prediction of potential crop yield is important in planning and policymaking for food security and agricultural economic return prediction [26,28,31]. This could include developing policies for improving agriculture productivity and sustainability [28]. In feeding a growing population in LMICs, agricultural production systems must strive to reduce the food production yield gap between current yields achieved by farmers and those potentially attainable in rainfed subsistence farming systems. In addressing this mismatch, the study by Hochman et al. [30] developed a model that integrated statistical yield and cropping area data, remotely sensed data, cropping system simulation, and GIS mapping to assess and map wheat yield gaps.

### 3.2.2. Soil Quality/Fertility Assessment

Soil quality assessment is critical for designing sustainable agricultural practices (optimal agricultural use) that can help bridge the current food production and demand gap in overcoming the food security problem. The availability of RS datasets and GIS spatial modelling techniques provides new opportunities for measuring/evaluating soil quality at different spatial scales [33,36]. Shokr et al. [35] developed a spatially-explicit soil quality model by combining soil's physical, chemical, and biological properties and integrating these with a digital elevation model and Sentinel-2 satellite image to produce digital soil maps. Abdelfattah and Kumar [32] describe the application GIS-enabled web-based soil information system that provides a descriptive, quantitative, and geospatial soil database in a simple interface. The system was applied to determine the sufficiency potential of soils for plant growing and management. Using GIS and RS technologies, Abdellatif et al. [38] developed a spatial model for the assessment of soil quality. His model combined four main soil quality indices (soil fertility index, soil physical index, soil chemical index, and geomorphological properties Index) and GIS ordinary kriging spatial interpolation to map the soil quality index. The application of these GIS-based models provides evidence-based ways to improve soil quality management. This would enable decision makers, policy formulators, land-use planners, and agriculturalists to efficiently manage soil resources to ensure the sustainable use of agricultural lands according to their potential [34,36,37]. Thus, assessing soil quality indicators would be important for sustainable agricultural policies and practices and in achieving food security.

### 3.2.3. Crop Mapping and Monitoring Decision Support Systems

In an era of unpredictable climate changes, agricultural crop monitoring analysis could help government policymakers and farmers plan and design cropping patterns that adapt to water availability. Agricultural monitoring systems integrate multiple geospatial data sets and cropping system models into computer algorithms to spatially compute and simulate optimum scenarios for site-specific conditions for crop production [42]. A crop monitoring system is developed by integrating geospatial data obtained by high-resolution remote sensing with a web GIS geoportal interface [41]. Santosh and Suresh [39] demonstrated the uniqueness of combining GIS and RS in a tool for crop selection and rotation analysis at the farm level to improve crop management decisions. Cropping patterns simulation is determined by irrigation water availability, which in turn is affected by changes in climate and irrigation water extraction policies. Wang et al. [16] combined GIS and irrigation water availability simulation models to analyze the cropping patterns based on the forecast of irrigation water availability. A GIS web-based crop mapping and monitoring decision support system at the farm level could help farmers to access information and take appropriate measures to improve crop production [39]. Such a system can have a wider application in supporting agronomic decision making including optimizing land and labor productivities, enhancing higher cropping intensities, and

producing better crop yield [40]. This can increase crop production and ensure better crop management, in the long run, and precision irrigation management.

### 3.2.4. Agricultural Drought Assessment

Using spatial datasets generated by satellite RS and GIS technologies offers very useful information for assessing and modelling agricultural drought-risk patterns, monitoring drought conditions, and producing drought vulnerability (risk) maps [48]. Hoque et al. [43] integrated geospatial techniques with fuzzy logic to develop a comprehensive spatial drought risk inventory model for operational drought management. This model successfully identified the spatial extents and distribution of agricultural drought risk. Sehgal and Dhakar [44] used GIS and high spatial resolution RS-derived indicators of crop sensitivity to develop a methodology that assessed and mapped, at a local scale, key biophysical factors contributing to agricultural drought vulnerability. The drought vulnerability maps could inform policymakers in formulating spatially explicit policies for drought mitigation and intervention strategies [45,46]. In addition, vulnerability maps could be used to indicate where socioeconomic development policy programs should be given priority [47].

### 3.2.5. Pest and Crop Disease Detection and Management

Several geospatial tools and techniques continue to be developed to aid farmers in crop disease control and management strategies. Several studies [49–52] provide practical application of satellite RS data and Geospatial techniques for sustainable crop disease detection and management. RS technology including Airborne and satellite imagery acquired during growing seasons has been used for early- and within-season detection, mapping of some crop diseases, the control of recurring diseases in future seasons, and assessing economic loss caused by frost damage [49]. Santoso et al. [50] used high-resolution QuickBird satellite imagery to effectively detect spatial patterns of oil palm plants infected by basal stem rot disease. They used six vegetation indices derived from visible and near-infrared bands satellite imagery to successfully discriminate between healthy and infected oil palms. Using precision agriculture technologies and remote sensed imagery, Yang [52] showed how site-specific fungicide application to disease-infested areas has been implemented for effective control of the disease. In the future, new approaches that apply geoinformation technologies in monitoring and management of pest and crop disease detection could reduce the effect of pesticides and herbicide chemicals on the environment.

### 3.2.6. Precision Agriculture

In precision agriculture, automated geospatial analysis and decision support algorithms can provide valuable scientific information to policymakers for better agriculture policy development. Precision agriculture practices, which employ integrated GIS, RS, and GPS technologies, have gained prominence in their ability to optimize crop production, facilitate site-specific crop management, and reduce the application of agrochemicals. Toscano et al. [58] demonstrated the usefulness of Sentinel-2 and Landsat-8 images to depict the within-field spatial variability of wheat yield, which is key for adopting precision farming techniques. This provided a potential alternative to traditional farming practices by improving site-specific management and agricultural productivity. García et al. [59] tested the performance of remote sensing drones as mobile gateways to provide a guide to the optimal drone parameters for successful Wi-Fi data transmission between sensor nodes and the gateway in precision agriculture systems. The study successfully demonstrated that drones (flying at the lowest velocity, at a height of 24 m, and with an antenna with 25 m of coverage) can be used as a remote sensing tool to gather the data from the nodes deployed on the fields for crop monitoring and management. This had the potential to increase the adoption of precision agriculture by even smallholder farmers. Segarra et al.'s [60] study specifically focused to understand the European Space Agency's twin Sentinel-2 satellites' features and their application in precision agriculture. Their study highlights that Sentinel-2 has dramatically increased the capabilities for agricultural moni-

toring and crop management, abiotic and biotic stresses detection, improved the estimation of crop yields, enhanced crop type classifications, and provided a variety of other useful applications in agriculture. All of these contribute to increasing the adoption of precision agriculture, which leads to more productive and sustainable agriculture management and environmental sustainability [61,62]. In precision agriculture, plantation-rows extraction using satellite image-based solutions is essential in crop harvesting, pest management, and plant grow-rate predictions. The study of Fareed and Rehman [55] used GIS and RS to design an automated method to extract plantation rows from a drone-based image point clouds-based digital surface model. The automatic plantation rows extraction can be used to quantify plantation-row damage assessment in precision agriculture.

### 3.2.7. Weed Management and Fertilizer Decision Support System

Accurate weed distribution mapping could greatly enhance efficiency in weed management and reduce weed damage, overhead costs of herbicide application, and the rationalization of fertilizers [57]. Dunaieva et al. [21] used GIS technologies to produce accurate weed distribution maps in rice farms. This information improved the efficiency of input application, thus reducing the consumption of inputs including herbicides, fungicides, and weeding labor costs. This in turn reduced the weed damage and crop production overhead costs. Xie et al. [56] demonstrated the application of GIS in the development of a GIS-based Fertilizer Decision Support System (FDSS) by integrating RS data, field surveys, and expert knowledge to develop a soil spatial database on the SuperMap platform for crop management systems. The application of FDSS in agricultural production had benefits, such as increasing fertilizer utilization efficiency, thus lowering production costs.

### 4. Obstacles to Applying GIS in Agriculture Policy and Practice

Generally, the use of GIS and RS technologies is not a panacea to successful evidence-based policy and practice and has its downside. The success of the geospatial technology application depends on its proper use, quality data, and considerable resources in its management. In countries that suffer low resources, such as LMICs, the cost of the technology and lack of appropriate skills jeopardize its wider use and adoption [63]. Simulating crop yield production is always challenging due to the variety of cropping systems and levels of technology used. Accurate crop yield gap assessment would require improvements in input data quality, including accurate weather parameters, better soil characterization, and spatially distributed land use data [28]. It would also demand the setting up of instrumented geo-referenced validation sites that provide comprehensive survey data to inform a continuous improvement cycle for yield gap assessment [30]. As such, future improvements in current remote sensing technology and the development of better-integrated cropping systems models would provide more accurate inputs for yield gap assessment.

In drought vulnerability assessment and mapping, most studies reported in the literature tended to use aggregated spatial data at higher spatial scales (national or regional level), but not at a finer scale (i.e., local level). Since the intensity of drought hazards is more felt and manifested at the local level, a detailed drought-risk mapping at a finer scale would require high resolution remote sensing and the use of locally contextual indicators to yield a full picture of vulnerability. This would have more relevance to policymakers whose intent is to formulate and implement mitigation interventions at the local level. With the prediction of more severe and frequent drought uncertainties and increasingly threats from climate change, drought-risk mapping that incorporates all the spatially explicit risk components would be a highly efficient contribution to drought-mitigating strategies. More skills and knowledge on the use of geospatial techniques for agricultural drought risk are needed.

In crop disease detection, challenges still exist in mapping them using airborne or satellite imagery. Although many crop diseases can be successfully detected and mapped using satellite imagery, each disease has its characteristics that would require different procedures for detection and management. According to Yang, [52] "some diseases are difficult to detect, especially when multiple biotic and abiotic conditions with similar

spectral characteristics exist within the same field" (pg. 531). Recurring diseases would require consistent historical imagery and spatial-temporal data, while emerging diseases are more difficult to detect. Yang argues that more advanced RS imaging sensors and image-processing techniques for differentiating diseases from other confounding factors are needed. In less-developed countries, very few farmers have the necessary skills required to use RS technologies in creating their prescription maps, in the implementation of disease management and in the site-specific fungicide application. More research is needed in the development of integrated geospatial analytical methodologies and tools for aiding farmers in the detection of different crop diseases.

Although precision agriculture technologies can aid in optimizing crops and facilitating agricultural management decisions in solving food insecurity challenges in LMICs, precision farming requires the adoption of geospatial technology and a large amount of high-resolution spatiotemporal data. A lack of skills to use GIS and RS in LMICs can be augmented by the dissemination and the transfer of practical geospatial technologies from developed countries [51]. However, considerable investments in ICT infrastructure are needed for the effective adaptation of precision agricultural approaches in LMICs.

Soil fertility assessment is considered one of the most important indicators of precision farming and for the sustainable use of agricultural lands according to their potential. This requires a comprehensive soil information system. However, according to Abdelfattah and Kumar [32], much of the world has very poor coverage of soil quality data. In LMICs, the fragmentation of agricultural land into small uneconomical plots and unsustainable farming practices is happening at a much higher rate. In such a rapidly changing environment, the potential of active remote sensors to determine soil quality requires further research.

Other obstacles to the use and adoption of GIS and RS in agriculture include a lack of commonly agreed data interoperability standards. Though there is increasing availability of spatial data usage in LMICs, many of these data are prone to error and are often collected and stored with different spatial units, formats, metadata, time, and space intervals. This makes some data unusable, prevents spatial data integration, and hinders a unified analysis of data, especially those collected from multiple sensors and platforms. A need exists on developing standardized guidelines for agriculture spatial data. Training for researchers, practitioners, and farmers on how to collect quality and accurate spatial data that can be usable in multi-platform systems is paramount. Developing spatial data repositories with better interoperability can enable data integration and improve the efficiency of data analyses. In this regard, crowdsourced data collection would be a promising contribution to developing cost-effective agri-spatial data repositories.

## 5. Conclusions

This paper has explored various ways GIS technology has been integrated into agriculture to improve agriculture decisions and policymaking. GIS and RS technologies present better methods for the analysis of spatial factors that affect agricultural production as compared to approaches where geographically explicit data are absent. If well exploited, the spatially integrated knowledge provided by GIS and RS can be applied to enhance agriculture policy and evidence-based interventions geared towards improving agriculture sustainability. Though GIS technologies provide a promising pathway for improving agronomic practices, they remain underexploited in many LMICs where a dire need for enhancing agriculture and food production practices is most needed. For LMIC governments and farmers to better exploit the benefit of GIS and RS technologies, there is a need for an increased level of awareness and potential use of spatial data related to agriculture. Further advances in geoinformatics techniques and computing infrastructure will allow a more collaborative framework amongst scientists, policymakers, researchers, extension personnel, crop consultants, and farm equipment and chemical dealers with practical guidelines for effective management of crop yield estimations, soil fertility, cropping pattern and monitoring, drought risks, and fertilizer and weed management.

In enhancing evidence-based agriculture policy, government and policymakers would require hard evidence that brings a clear understanding of the complexity and interconnectedness of factors affecting agriculture productivity. This would in turn enable designing concrete intervention strategies. Additionally, a broad spectrum of stakeholders and practitioners in the agriculture sector would need location-specific agricultural data in implementing a wide array of decisions that improve agricultural production potential. Equally, smallholder farmers would require synthesized information to empower them not only to make evidence-informed decisions but also to implement practicable activities that increase agricultural productivity. This raises the demand for GIS integration in agriculture policy formulation and implementation.

GIS and RS technologies provide a big potential in the assessment, storage, processing, and production of agriculture data. The data could be useful in precision agriculture, site-specific farming, and disease detection, among others, all geared toward improving agriculture food production and food security issues. Unfortunately, the lack of quality spatial data in many local governments has continued to undermine decision-making, policy formulation and the effectiveness in their implementation. Where such data exists, there is a generally lack of skills on the use of GIS and RS spatial analytical techniques. Achieving spatially integrated agriculture policies demands comprehensive, up-to-date spatial datasets and better methods that combine and analyze complex data from various sources to produce useful information. This would necessitate national as well as local governments to adopt methods, strategies, and techniques that facilitate the collection and analysis of diverse agricultural datasets in providing comprehensive insights to policymakers, planners, farmers, and a broad spectrum of stakeholders in the agriculture sector. GIS thus provides a promising pathway for the acquisition of comprehensive, up-to-date spatial databases and better spatial analysis methods that are capable of analyzing complex data to produce useful information. If properly adopted and implemented, GIS can enhance the spatial decision support system in improving the efficiency and effectiveness of agriculture policy formulation and planning. Nonetheless, policy change can guide and catalyze actions but requires public and political will to actualize it. Thus, the adoption of GIS technology in policymaking would require local government to commit public funds to set up the required software, hardware, supportive infrastructure, and training of staff to use them. Future studies can focus on how GIS and RS technology could promote a collaborative framework amongst scientists, policymakers, researchers, extension agriculture officers in promoting sustainable, and climate-smart farming practices, especially in LMICs.

## 6. Limitations of the Study

The results of this study are purely based on the 40 articles spanning 10 years (2010–2021) and the learning obtained from them. As a result, we might not have included some significant research papers published in earlier years. However, the purpose of this review was to analyze the most recent trends and relevant publications in the application of GIS in agriculture. For this reason, we argue that the 40 papers are comprehensive, and it is unlikely that the content of previously published papers would have substantially altered our findings. Furthermore, the selection criteria only included peer-reviewed papers. However, reflections on GIS methodologies are sometimes published in books or grey literature since they have more space for in-depth reflections. To reduce this limitation, we formulated our search string to include a broad range of the most relevant terms of interest in this study. In addition, we performed the search in the largest indexed databases of SCOPUS, BASE, CORE, and Clarivate. To account for significant papers that might have not been indexed in these databases, we also included the search results from Google Scholar. Notwithstanding these few limitations, the insights provided in this review provide valuable information and knowledge on GIS and RS application in enhancing evidence-based policy interventions for enhancing agriculture sustainability, as well as identifying barriers to their application in the LMIC context.

**Author Contributions:** M.M.; conceptualization, methodology, formal analysis, and, writing—original draft preparation. J.E.W.B. and B.G.J.S.S.; supervision, writing—review and editing, study design and funding acquisition. All authors have read and agreed to the published version of the manuscript.

**Funding:** This research was funded by NUFFIC, grant number NICHE-KEN-284, and managed by Vrije University Amsterdam. The Spatial Planning and Agribusiness Development (SPADE) project is a collaboration project between Maseno University Kenya, and Vrije Universiteit Amsterdam, implemented in Kenya.

**Conflicts of Interest:** The authors declare no conflict of interest.

## Appendix A

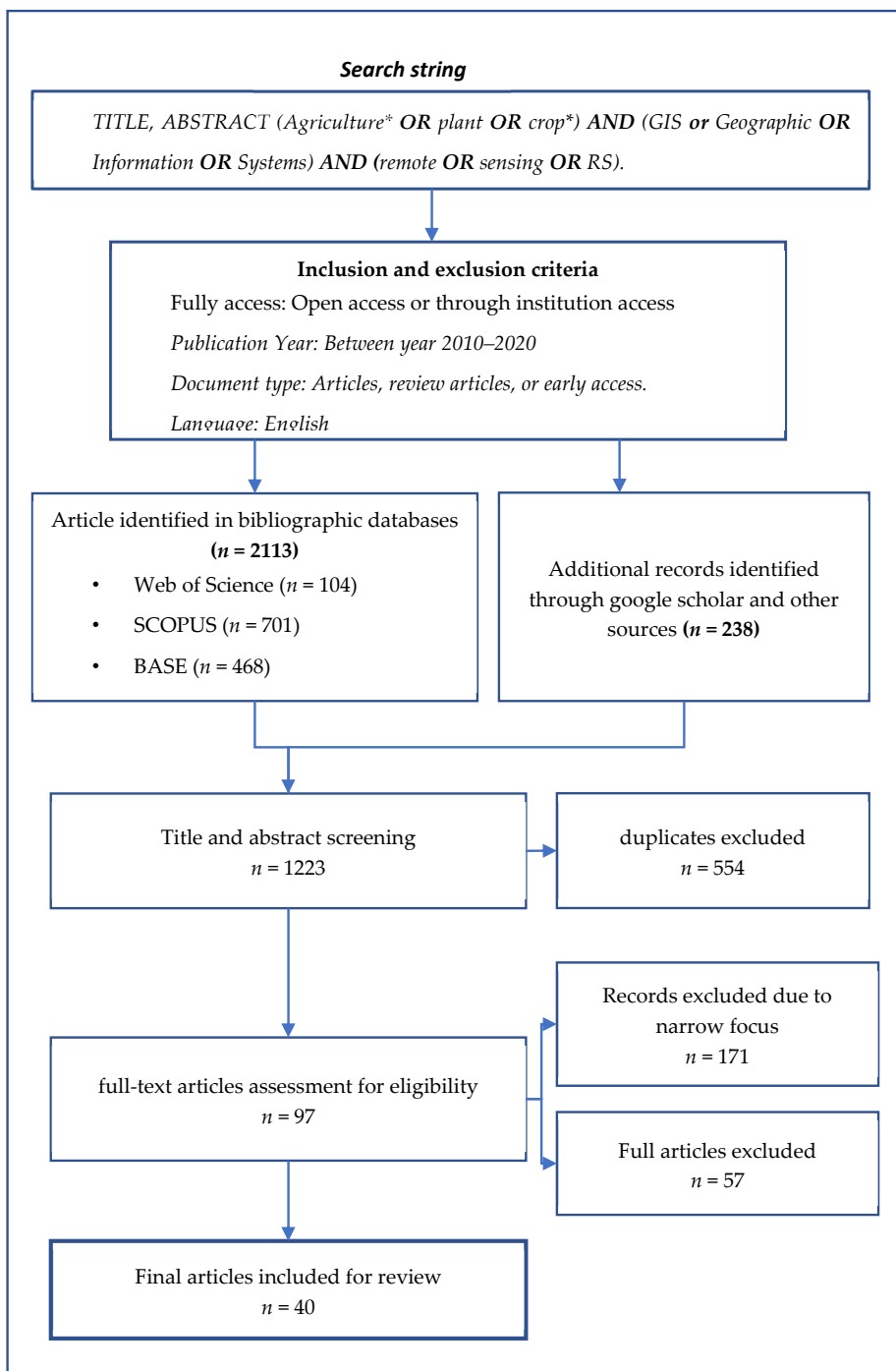

**Figure A1.** Flow diagram of the screening process.

**Table A1.** Search syntax.

| CORE database search string: (date of search: 25.10.2021): 68 articles found |
|---|
| **title:((Agriculture, AND GIS, AND Remote AND sensing,)) AND year: [2010 TO 2021]** |
| Web of Science query (date of search: 26.10.2021): 104 articles found |
| https://www.webofscience.com/wos/woscc/summary/79c860ee-207f-46ca-8db2-f70df81462ed-0eff1eca/relevance/1 |
| Search refined by: **Publication Years**: Between 2021 and 2010: **Document Types**: Articles or Review Articles or Early Access: **Languages**: English: **Open Access:** All Open Access or Gold or Gold-Hybrid or Green Published or Free to Read: **Document Types:** Articles **Web of Science Categories:** Remote Sensing or Geosciences Multidisciplinary or Green Sustainable Science Technology or Multidisciplinary Sciences or Agronomy or Ecology or Geography Physical or Plant Sciences or Agriculture Multidisciplinary or Computer Science Information Systems or Soil Science or Agricultural Engineering or Agricultural Economics Policy or Food Science Technology or Horticulture or Nutrition Dietetics or Environmental Sciences or Environmental Studies. |
| SCOPUS search query (date of search: 26.10.2021) |
| TITLE-ABS-KEY (*gis,* AND *agriculture,* AND *remote* AND *sensing,* OR *rs* OR *geographic* OR *information* OR *systems*) AND (LIMIT-TO (PUBYEAR, *2022*) OR LIMIT-TO (PUBYEAR, *2021*) OR LIMIT-TO (PUBYEAR, *2020*) OR LIMIT-TO (PUBYEAR, *2019*) OR LIMIT-TO (PUBYEAR, *2018*) OR LIMIT-TO (PUBYEAR, *2017*) OR LIMIT-TO (PUBYEAR, *2016*) OR LIMIT-TO (PUBYEAR, *2015*) OR LIMIT-TO (PUBYEAR, *2014*) OR LIMIT-TO (PUBYEAR, *2013*) OR LIMIT-TO (PUBYEAR, *2012*) OR LIMIT-TO (PUBYEAR, *2011*) OR LIMIT-TO (PUBYEAR, *2010*)) AND (LIMIT-TO (DOCTYPE, *"ar"*)) AND (LIMIT-TO (LANGUAGE, *"English"*)) AND (LIMIT-TO (EXACTKEYWORD, *"Remote Sensing"*) OR LIMIT-TO (EXACTKEYWORD, *"GIS"*) OR LIMIT-TO (EXACTKEYWORD, *"Agriculture"*) OR LIMIT-TO (EXACTKEYWORD, *"Geographic Information Systems"*) OR LIMIT-TO (EXACTKEYWORD, *"Agricultural Land"*) OR LIMIT-TO (EXACTKEYWORD, *"Geographic Information System"*) OR LIMIT-TO (EXACTKEYWORD, *"Remote Sensing And GIS"*) OR LIMIT-TO (EXACTKEYWORD, *"Remote Sensing Technology"*) OR LIMIT-TO (EXACTKEYWORD, *"Spatial Analysis"*) OR LIMIT-TO (EXACTKEYWORD, *"Agricultural Robots"*) OR LIMIT-TO (EXACTKEYWORD, *"Soils"*) OR LIMIT-TO (EXACTKEYWORD, *"Sustainability"*) OR LIMIT-TO (EXACTKEYWORD, *"Decision Making"*) OR LIMIT-TO (EXACTKEYWORD, *"Soil"*) OR LIMIT-TO (EXACTKEYWORD, *"Precision Agriculture"*) OR LIMIT-TO (EXACTKEYWORD, *"Prediction"*) OR LIMIT-TO (EXACTKEYWORD, *"Crop Production"*) OR LIMIT-TO (EXACTKEYWORD, *"Agricultural Production"*) OR LIMIT-TO (EXACTKEYWORD, *"Crop Yield"*) OR LIMIT-TO (EXACTKEYWORD, *"Agricultural Development"*) OR LIMIT-TO (EXACTKEYWORD, *"Drought"*) OR LIMIT-TO (EXACTKEYWORD, *"Irrigation (agriculture)"*) OR LIMIT-TO (EXACTKEYWORD, *"Geographic Information System (GIS)"*) OR LIMIT-TO (EXACTKEYWORD, *"Alternative Agriculture"*) OR LIMIT-TO (EXACTKEYWORD, *"Food Security"*)) |

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
