# Peer review of "Application of GIS in Agriculture in Promoting Evidence-Informed Decision Making for Improving Agriculture Sustainability: A Systematic Review"

_sustainability, doi:10.3390/su14169974_

Round 1
Reviewer 1 Report
This paper focused on previous researches to explore how GIS and RS affect agriculture policy in Africa of recent 10 years. To synthesize existing evidence on GIS and RS application in agriculture in enhancing evidence-informed policy for improving agriculture sustainability and identify obstacles to their application, particularly in low- and middle-income countries, this paper conducted systematic searches in lots of research databases and picked out 40 articles for profound research.
Still, there’s some specific problems need to be solved or reconstructed. If it’s done, this paper would be important for agriculture development.

Author Response
Dear reviewer #1,
We thank you for your time to review our manuscript and are indebted by the insightful comments and suggestions you have given us in improving our submitted manuscript. Your comments have improved our manuscript and enriched its contents. We believe that the paper is now better packaged, more coherent, and relevant to the readers and subscribers of the Sustainability journal
In the attached table, we detail the changes made to the paper and the relevant line numbers the changes have been made.
With best regards,
Mathenge M.
(Corresponding author).

Reviewer 2 Report
The review "Application of GIS in agriculture in promoting spatially inte- 2 grated policies for improving agriculture sustainability: A sys- 3 tematic review" is good but should be improved before publication
Kindly read through and give a professional English editor to correct the flow of thought
The appendix 1A needs to be drawn properly and Legibly
The references should be properly formatted to MDPI Guidelines. Especially the Ref section. full of improper referencing
Author Response
Dear reviewer #2,
We thank you for your time to review our manuscript and are indebted by the insightful comments and suggestions you have given us in improving our submitted manuscript. Your comments have improved our manuscript and enriched its contents. We believe that the paper is now better packaged, more coherent, and relevant to the readers and subscribers of the Sustainability journal
In the attached table, we detail the changes made to the paper and the relevant line numbers the changes have been made.
With best regards,
Mathenge M.
(Corresponding author).

Reviewer 3 Report
The article entitled “Application of GIS in agriculture in promoting spatially integrated policies for improving agriculture sustainability: A systematic review” is written very well and according to the scope of the journal. It is publishable after addressing some issues.
1. I suggest you to start the abstract directly from the main objectives of the study.
2. In the first paragraph of the introduction, after the sentence “However, the growing concerns are that the agricultural food production systems are unable to match the high demand, especially in poor nations, causing an intensifying level of food insecurities [2]”, you must have to add the given sentence with given study as “However, the growing concerns are that the agricultural food production systems are unable to match the high demand, especially in poor nations, causing an intensifying level of food insecurities [2]. The inefficiency of the food production system is also one of the reasons for food insecurity [1]”.
[1] Application of an artificial neural network to optimise energy inputs: An energy-and cost-saving strategy for commercial poultry farms. Energy. Volume 244, 123169.
3. From lines 48 to 51, the given sentence should be updated with the given study as “These include planting climate resilient crops, high-yielding crop varieties, crop yield forecasting, integrated pest management, as well as integrating biodiversity solutions in sustainable food production systems [2]”.
[2] Extreme weather events risk to crop-production and the adaptation of innovative management strategies to mitigate the risk: A retrospective survey of rural Punjab, Pakistan. Technovation. Volume 117
4. I recommend you add the structure of the article at the end of the introduction.
5. What are the main contributions of the study? You must have to add it at the end of the introduction.
4. From the lines 334 to 335, you must have to update the sentence with the given studies as “All of these contribute to increasing the adoption of precision agriculture, which leads to more productive and sustainable agriculture management, and environmental sustainability [3,4]”
[3] Understanding farmers’ intention and willingness to install renewable energy technology: A solution to reduce the environmental emissions of agriculture. Applied Energy. Volume 309, 118459.
[4] Understanding cognitive and socio-psychological factors determining farmers’ intentions to use improved grassland: Implications of land use policy for sustainable pasture production. Land Use Policy, 102, 105250
5. Limitations of the study should write after the conclusion section.
6. Please suggest some future studies at the end of the conclusion section.
Best of luck!
Author Response
Dear reviewer #3,
We thank you for your time to review our manuscript and are indebted by the insightful comments and suggestions you have given us in improving our submitted manuscript. Your comments have improved our manuscript and enriched its contents. We believe that the paper is now better packaged, more coherent, and relevant to the readers and subscribers of the Sustainability journal
In the attached table, we detail the changes made to the paper and the relevant line numbers the changes have been made.
With best regards,
Mathenge M.
(Corresponding author).

Round 2
Reviewer 1 Report
The paper has been revised better. The revised paper has better innovation. I recommend that the paper be revised for publication.